# Structural equation model of psychological constructs of transtheoretical model, motives for physical activity, and amount of physical activity among people with type 2 diabetes mellitus in Malaysia

**Aizuddin Hidrus**[1,2]*, **Yee Cheng Kueh**[1,3]*, **Bachok Norsa'adah**[1]*, **YoungHo Kim**[4], **Yu-Kai Chang**[5,6], **Garry Kuan**[7,8]*

1 Biostatistics and Research Methodology Unit, School of Medical Sciences, Universiti Sains Malaysia, Kubang Kerian, Kelantan, Malaysia, 2 Public Health Medicine Department, Faculty of Medicine and Health Science, Universiti Malaysia Sabah, Kota Kinabalu, Sabah, Malaysia, 3 Hospital Universiti Sains Malaysia, Kubang Kerian, Kelantan, Malaysia, 4 Department of Sport Sciences, Seoul National University of Science and Technology, Nowon-gu, Seoul, Korea, 5 Department of Physical Education and Sport Sciences, National Taiwan Normal University, Taipei, Taiwan, 6 Institute for Research Excellence in Learning Science, National Taiwan Normal University, Taipei, Taiwan, 7 Exercise and Sports Science, School of Health Sciences, Universiti Sains Malaysia, Kubang Kerian, Kelantan, Malaysia, 8 Department of Life Sciences, Brunel University London, London, United Kingdom

* yckueh@usm.my (YCK); norsaadah@usm.my (BN); garry@usm.my (GK); aizuddin88@ums.edu.my (AH)

## Abstract

### Background

This study aimed determine the structural relationship between psychological constructs of the transtheoretical model (TTM; processes of change, decisional balance, and exercise self-efficacy), motives for physical activity (PA), and amount of PA among Malaysians with type 2 diabetes mellitus (T2DM).

### Method

All participants were recruited from the Hospital Universiti Sains Malaysia using a cross-sectional study design with purposive) sampling method. A total of 331 participants were recruited for the present study. Before participation in the study, they were informed that participation in the study was totally voluntary. Those who agreed to participate voluntarily were required to complete the self-administered questionnaire set, which included the processes of change, decisional balance, exercise self-efficacy, physical activity and leisure motivation, and international physical activity questionnaires. Data analysis of structural equation modeling was performed using Mplus 8.

### Results

From the 331 participants, most of whom were male (52%) and Malay (89.4%), with a mean age of 62.6 years (standard deviation = 10.29). The final structural equation model fit the

**Data Availability Statement:** All relevant data are within the paper and its Supporting Information files.

**Funding:** Yee Cheng Kueh has received the Ministry of Higher Education Malaysia for Fundamental Research Grant Scheme (FRGS) with Project Code: FRGS/1/2020/SKK06/USM/03/1 (Funder website: https://www.mohe.gov.my/en/services/research/mygrants). The funders had no role in study design, data collection and analysis, decision to publish, or preparation of the manuscript.

**Competing interests:** The authors have declared that no competing interests exist.

data well based on several fit indices [Root Mean Square Error of Approximation (RMSEA) = 0.059, Comparative Fit Index (CFI) = 0.953, Tucker-Lewis Index (TLI) = 0.925, Standardized Root Mean Square Residual (SRMR) = 0.031]. A total of 16 significant path relationships linked between the TTM, motives for PA, and amount of PA.

## Conclusion

The pros of decisional balance, others' expectations, and psychological condition were constructs that directly affected PA, whereas the other constructs had a significant indirect relationship with the amount of PA. A positive mindset is crucial in deciding a behavioral change toward an active lifestyle in people with T2DM.

## Introduction

In the last century, physical activity (PA) has become a key pillar of healthy lifestyles. A physicist named Hippocrates has admitted since the fifth century BC that "all parts of the body, if used in moderation and exercised in labors to which each is accustomed, become thereby healthy and well developed and age slowly; but if they are unused and left idle, they become liable to disease, defective in growth and age quickly" [1]. The discovery of population PA levels began in the early 1970s and 1980s [2]. Physical inactivity was identified as one of the most important risk factors for heart disease [3]. Since then, fear of physical inactivity has become one of the major health problems. Doctors and researchers have therefore initiated numerous studies and research on the association between reduced PA and diseases [4–7].

For decades, other than diet and medication, PA has been considered as a foundation of diabetic management [8]. Developing and maintaining patients' motivation toward PA is crucial, as it may be an approach for the management and treatment of the disease. The American Diabetes Association [9] stated that "The possible benefits of PA for the patient with type 2 diabetes mellitus (T2DM) are substantial, and recent studies strengthen the importance of long-term PA programs for the treatment and prevention of this common metabolic abnormality and its complications". This statement shows that not only regular medications but also PA can play a major role in improving T2DM patients' conditions. Researchers and clinicians face two main challenges: how to get people to practice and how to help active people maintain their practice [10]. Both challenges are related to people's psychological perspectives and motivation toward PA.

Psychologically, the transtheoretical model (TTM) is the model suggested that behavior change, such as quitting tobacco, should be viewed more as a continuum than binary: the shift from risky to healthy behavior [11]. Other than smoking, it also been used for overweight and diet problem [12] and those with physically inactive behavior [13]. Initially developed since the 1970s and 1980s by Prochaska, DiClemente, and colleagues, the TTM, also known as the stage of change model, finally matured in the 1990s [14]. TTM represents the dynamic idea of conducting changes for wellbeing, including exercising, and perceives that people should regularly make a few endeavors to conduct change before they achieve their goals [15]. TTM consists of four core constructs: 1. the six stages of exercise behavior change (i.e., pre-contemplation, contemplation, preparation, action, maintenance, and relapse [16]); 2. the psychological constructs, which consist of processes of change (overt and covert activities that individuals utilize to modify their behavior [15]); 3. decisional balance (involves the perceived "pros" (advantages) and "cons" (disadvantages) of continuing a current behavior or adopting a

new behavior [17]; and 4. exercise self-efficacy (how well one can execute courses of action required to deal with prospective situations [18]).

Motivation plays an important role in participation in physical/sporting activities [19]. It is one of the essential components in the psychological process of individuals in deciding their participation in physical activities. The same is true for exercise and PA, where motivation plays a major role not only in promoting involvement in PA but also in maintaining this involvement [20, 21]. Various types of motivation have been shown to influence people's efforts during exercise sessions and their intentions to continue exercising [22]. Developed by Rogers and Morris [23], the physical activity and leisure motivations scale (PALMS) is an instrument with 40 items that is used to measure an individual's motivation for PA. These 40 items are classified into eight different factors or motives: enjoyment, mastery, competition/ ego, appearance, affiliation, others' expectations, psychological condition, and physical condition.

Yet, the researchers found a lack of research about the motives for PA and TTM psychological variables in Malaysian people with T2DM. We believe that the application and exposure of these constructs to Malaysian people with T2DM could produce a positive effect on their control of blood sugar levels. Hence, the objective of this study was to determine the structural relationships between TTM psychological constructs, motives for PA, and amount of PA among Malaysian people with T2DM. Moreover, we examined the direct and indirect relationships between those constructs. The initial SEM model of the present study was developed by adapting a study conducted by Liu et al [24]. Liu examined the relationship between TTM of behavior change with amount of PA among university students in Malaysia. In the present study, the SEM analysis included the TTM's psychological variables with additional eight motives for PA in the hypothesized SEM model. Fig 1 shows the hypothesized proposed structural relationships of our study.

## Materials and methods

### Study design and participants

We used a cross-sectional study design in order to collect information of the study variables. People with T2DM were recruited via non-probability (purposive) sampling method from the Hospital Universiti Sains Malaysia (USM). They were recruited from two clinics (Family Care Clinic and Diabetes Specialist Clinic) and four wards (Male and Female Medical wards and

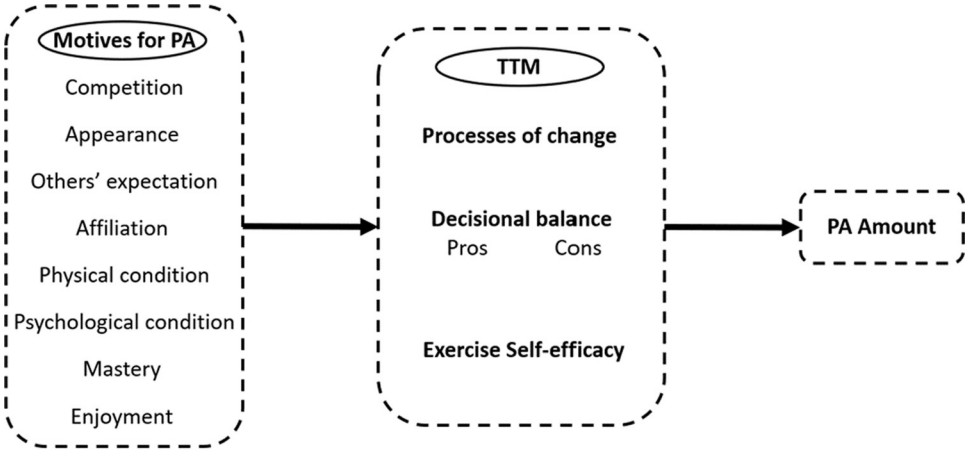

**Fig 1. Hypothesized proposed relationships of TTM, motives for PA, and amount of PA.**

Male and Female Surgical wards) where most of the people with T2DM are reviewed and treated. They were informed about the study objective and the requirement of completing the set of questionnaires for participation in this study. The participants were required to fulfill the following inclusion criteria: Malaysian aged 18 years and above who was clinically diagnosed with T2DM for at least one year, could read and understand Malay language, agreed to the information to participants' form that was explained by the researcher. Whereby, any mental disorder was the exclusion criteria of the study. We managed to recruit a total of 331 Malaysians with T2DM from Hospital USM to participated in this study.

## Sample size determination

As recommended by Kline [25], sample size of 200 may be adequate for an SEM model. Yet, the sample size is depend on the complexity of the model. With Mplus 8, the facility of using monte carlo simulation is provided to estimate the sample size for hypothesized SEM model [26]. Providing the power for each regression coefficient (path relationship between the study variables), lists of sample size (200 to 320) were entered during the simulation. For this study, 0.80 was the targeted minimum power. Based on previous study on TTM's variables [24], 0.17 was the estimated standardised regression coefficient. From the simulation study, 200 sample size would achieve minimum power of 0.63, whereby sample size of 320 would achieve minimum power of 0.84. Hence, a total sample size of 331 in the present study is considered acceptable.

## Data collection

Data collection was performed using a self-administered questionnaire from the people with T2DM in Hospital USM. After the questionnaire was fully checked and prepared, participants' recruitment was carried out from the clinics and wards where people with T2DM attended and warded in Hospital USM. They were approached by two ways, first was by directly approaching them while they were waiting for their turn number to be called for the regular clinic review, or while they were on their bed for those who were admitted in wards. They were briefly explained about the study and invited to participate if they are interested.

The second way was through posters that have been patched on the signboard of the clinics and wards. The poster displays a brief explanation about the study and details of the person in charge (the researcher). Those who were willing to participate and eligible were then explained in detail about the study and required to signed consent form before answering the questionnaire.

## Measures and materials

**Processes of change scale.** The processes of change (POC) scale is a 30-item questionnaire initially developed by Nigg and colleagues [27] that was applied to eligible students to assemble information. Participants were required to answer the questionnaire on a five-point Likert scale, from "never" to "repeatedly". The questionnaire has two types of factor order: first-order factors, 1. consciousness raising, 2. dramatic relief, 3. environmental re-evaluation, 4. self-re-evaluation, 5. social liberation, 6. counter conditioning, 7. helping the relationship, 8. reinforcement management, 9. self-liberation, and 10. stimulus control, and second-order factors, 1. cognitive processes that formed of the numbers 1 to 5 of the first-order factors, and 2. behavior processes with the remaining five of the first-order factors. In the present study, the researchers adopted the second-order factor version for the analysis, as it is normally applied in a research study [26]. The translated Malay version of the POC scale (POC-M), which was validated by Hidrus et al. [28], was used. Model fit indices of the POC-M showed an acceptable

range value of comparative fit index (CFI) = 0.922, Tucker-Lewis index (TLI) = 0.911, root mean square error of approximation (RMSEA) = 0.048, and standardized root mean square residual (SRMR) = 0.064. POC-M also showed an acceptable reliability based on the value of composite reliability, 0.856 for cognitive processes, and 0.752 for behavioral processes.

**Decisional balance scale.** The decisional balance (DB) scale is a questionnaire that consists of 10 items and was initially developed by Plotnikoff et al. [17]. Similar to the POC scale, it also uses a five-point Likert scale, from 1 (not at all confident) to 5 (extremely confident). The DB scale consists of two factors: pros, which represent the positive aspects of an individual's behavioral changes, and cons, which represent the negative aspects. The researchers adopted the translated Malay version of the DB scale (DB-M) that has been validated by Kuan et al. [29]. The DB-M produced good results of model fit indices, with CFI = 0.979, TLI = 0.969, RMSEA = 0.047, and SRMR = 0.037.

**Exercise self-efficacy scale.** The original English version of the exercise self-efficacy (ESE) scale was developed by Bandura [30], with a single factor of exercise self-efficacy. The scale was translated into the Korean language and revised into three factors (internal feelings with seven items, competing demands with five items, and situational with six items) and 18 items that were adopted by Kim [15], Shin et al. [31], and Kosma et al. [32] in their studies. The scale was developed with a five-point Likert scale ranging from "1 = cannot do" to "5 = certain can do". The Malay version of the scale had been validated among 1605 university undergraduates [33]. Sabo et al. [33] tested two initial models (single- and three-factor models), which did not produce good fit indices. Through some modifications to the models, goodness of fit indices were yielded for both the single-factor model (RMSEA = 0.059, CFI = 0.939, TLI = 0.922, SRMR = 0.049) and the three-factor model (RMSEA = 0.066, CFI = 0.924, TLI = 0.903, SRMR = 0.051). The composite reliability for the single-factor model was 0.886, whereas that for the three-factor model ranged from 0.670 to 0.854.

Due to the discrepancy of the results found in the previous studies in regard to the factors of the structure of the ESE scale in this study, Hidrus et al. [34] conducted another validation study on the Malay version of ESE (ESE-M) among people with T2DM in Malaysia. Results showed that single-factor ESE-M (RMSEA = 0.054, CFI = 0.952, TLI = 0.938, SRMR = 0.044) was more suitable than the three-factor ESE-M to be adopted among people with T2DM. Therefore, we decided to treat ESE as a single factor for the analysis in this study.

**Physical activity and leisure motives scale.** The physical activity and leisure motives scale (PALMS) measures motives for participating in PA and leisure [23]. The PALMS questionnaire used in this study consisted of 40 items measuring the aspects of mastery, physical condition, psychological condition, affiliation, appearance, enjoyment, competition/ego, and others' expectations. Each factor consisted of five items, which were scored on a five-point Likert scale rated from 1 (*strongly disagree*) to 5 (*strongly agree*). The PALMS has shown good reliability with 4-week test-retest correlations ranging from .78 to .94 [35]. Internal consistency for all eight factors was good, ranging from 0.78 to 0.82, as measured based on Cronbach's alpha. Molanorouzi et al. [36] examined the validity of PALMS using CFA and reported that the eight factors in the 40-item scale had acceptable goodness of fit. The Malay version of PALMS (PALMS-M) was used in this study. The validity and reliability of the PALMS-M were found satisfactory when tested among 634 Malaysian undergraduate students [36]. The validation was based on CFA, with a majority of the fit indices achieving the threshold fit values (RMSEA = 0.041, CFI = 0.911, TLI = 0.901, SRMR = 0.052). The reliability based on composite reliability ranged from 0.648 to 0.846.

**International physical activity questionnaire.** The international physical activity questionnaire (IPAQ, short version) includes seven questions that measure the amount of PA based on four types of activities: vigorous, moderate, walking, and sitting. Continuous scoring

for IPAQ was applied for the present study. It is based on the following metabolic equivalent of task (MET) values: walking = 3.3 METs, moderate activity = 4.0 METs, and vigorous activity = 8.0 METs, which are expressed as MET-min per week: MET level × minutes of activity × events per week. A reliable and validated Malay version of IPAQ (IPAQ-M) was established by Chu and Moy [37]. Criterion validity was carried out between the IPAQ-M and a seven-day PA log (PA-log). The validity of the overall classification systems was tested and almost perfectly agreed (κ = 0.89; 95% CI = 0.79–0.98). The IPAQ-M has an interclass correlation coefficient ranging from 0.54 to 0.86, whereas the Spearman correlation coefficient ranged between 0.67 and 0.98. Based on these results, Chu and Moy concluded that the self-administered IPAQ-M is reliable and valid for adoption among Malaysian adults. The total score of IPAQ-M was used to measure the amount of PA.

## Ethical considerations

This study was approved by the Human Research Ethics Committee of USM (USM/JEPeM/ 18040201) and was conducted based on the guidelines of the International Declaration of Helsinki. The participants who volunteered and agreed to participate were given a research information sheet that contained important information, such as the objective of the study, study procedures, and the possible risks and benefits to the participants. All participants were initially required to complete consent forms before the study began as an indication that they volunteered and were willing to participate in this study. Then, they were required to fill in the consent form and took approximately 30–40 minutes to complete the questionnaire.

## Data analysis

Statistical analyses were performed using Mplus 8. Numerical variable data were presented as mean and standard deviation (SD), whereby frequencies and percentages were used to express categorical variable data for descriptive information. Structural equation modeling (SEM) was performed to assess the structural relationship of TTM psychological constructs, motives for PA, and PA amount. The estimator of maximum likelihood robust (MLR) by Yuan and Bentler [38] was used in the SEM analysis. The evaluation of the model's fitness was based on several model fit indices: CFI > 0.95, TLI > 0.95, RMSEA < 0.07, and SRMR < 0.08 [39].

The initial hypothesized model included five study variables: POC, DB, ESE, motives for PA, and amount of PA. We hypothesized the presence of inter-relationships between TTM of behavior changes, motives for PA, and amount of PA among the participants with T2DM. The hypothesized path relationships that were not significant were removed iteratively from the model. New pathways, as suggested by the modification index, were added to the model one by one after discussion among the researchers. After the final best fit model was obtained, indirect relationships related to PA were examined. The standardized indirect effect and the p-value produced by Mplus 8 were reported.

## Results

### Participants

The demographic details of 331 participants are shown in Table 1. Among the 331 participants, 172 (52%) were male, and 159 (48%) were female. The mean age of the participants was 62.6 years (standard deviation (*SD)* = 10.29 years), with an age range of 22 to 94 years. The majority of the participants were Malay (*n* = 296: 89.4%), and the remaining 10.6% were Chinese, Indian, or other ethnicities. As for education background, most of the participants were at the secondary school level (*n* = 158: 47.7%). Out of the 331 participants, 139 (42%) were

**Table 1. Demographic characteristics and medical variables of people with T2DM in Hospital USM (*n* = 331).**

| Characteristics | Frequencies | Percentage | Mean (*SD*) |
|---|---|---|---|
| **Gender** | | | |
| Male | 172 | 52.0 | |
| Female | 159 | 48.0 | |
| **Age (years)** | | | 62.6 (10.29) |
| **Age group** | | | |
| **21–40 years old** | 9 | 2.7% | |
| **41–60 years old** | 124 | 37.5% | |
| **61–80 years old** | 189 | 57.1% | |
| **81–100 years old** | 9 | 2.7% | |
| **Ethnicity** | | | |
| Malay | 296 | 89.4 | |
| Chinese | 25 | 7.6 | |
| Indian | 6 | 1.8 | |
| Others | 4 | 1.2 | |
| **Education background** | | | |
| Primary | 87 | 26.3 | |
| Secondary | 158 | 47.7 | |
| Diploma | 60 | 18.1 | |
| Bachelor degree | 26 | 7.9 | |
| **Occupation** | | | |
| Working/Business | 92 | 27.9 | |
| Pensioners | 139 | 42.0 | |
| Not working/Housewife | 100 | 30.1 | |
| **Diabetic period** | | | |
| Less than 5 years | 38 | 11.5 | |
| $\geq$ 5 years | 32 | 9.7 | |
| $\geq$ 10 years | 83 | 25.1 | |
| $\geq$ 20 years | 178 | 53.8 | |
| **HbA1c (mmols/mol) (n = 279)** | | | 76.90 (1.33) |
| **BMI (kg/m²)** | | | 27.28 (0.28) |

Note. HbA1c = glycated hemoglobin, BMI = body mass index

pensioners, 92 (27.9%) were still working/had a business, and the remaining were not working/were housewives (*n* = 100: 30.1%).

## Structural model

The initial structural model has 36 hypothesized path relationships. Table 2 shows the results of the goodness of fit indices for initial model, model two, and final model. The results

**Table 2. Goodness of fit indices for hypothesized structural model.**

| Model | CFI | TLI | SRMR | RMSEA (90% CI) | RMSEA (*p*-value) |
|---|---|---|---|---|---|
| **Initial model (Model-1)** | 0.822 | 0.552 | 0.053 | 0.130 (0.111, 0.149) | <0.001 |
| **Model-2** | 0.849 | 0.759 | 0.075 | 0.106 (0.089, 0.123) | <0.001 |
| **Model-3** | 0.953 | 0.925 | 0.031 | 0.059 (0.040, 0.078) | 0.209 |

CFI = comparative fit index, TLI = Tucker-Lewis index, SRMR = standardized root mean square residual, RMSEA = root mean square error of approximation, CI = confidence interval

**Table 3. Hypothesized path relationships in modified proposed model.**

| Pathways | Standardized regression coefficient, β (95% CI) | Critical Ratios | p-value |
|---|---|---|---|
| POC ← Competition | 0.137(0.013, 0.260) | 2.169 | 0.030 |
| Cons ← Competition | -0.220(-0.348, -0.091) | -3.348 | 0.001 |
| ESE ← Competition | 0.158(0.007, 0.309) | 2.054 | 0.040 |
| ESE ← Others' expectation | 0.140(0.041, 0.239) | 2.766 | 0.006 |
| Pros ← Physical condition | 0.229(0.092, 0.365) | 3.287 | 0.001 |
| Pros ← Psychological condition | 0.277(0.126, 0.427) | 3.604 | <0.001 |
| Cons ← Psychological condition | -0.227(-0.337, -0.118) | -4.075 | <0.001 |
| POC ← Mastery | 0.251(0.116, 0.386) | 3.638 | <0.001 |
| ESE ← Mastery | 0.344(0.207, 0.481) | 4.909 | <0.001 |
| POC ← Enjoyment | 0.146(0.004, 0.287) | 2.021 | 0.043 |
| PA ← Pros | 0.223(0.073, 0.373) | 2.912 | 0.004 |
| POC ← Pros | 0.269 (0.103, 0.435) | 3.176 | 0.001 |
| POC ← ESE | 0.315 (0.214, 0.416) | 6.110 | <0.001 |
| Pros ← POC | 0.376 (0.237, 0.516) | 5.286 | <0.001 |
| PA ← Others' expectation | 0.180(0.054, 0.306) | 2.794 | 0.005 |
| PA ← Psychological condition | 0.188(0.056, 0.320) | 2.788 | 0.005 |

indicated that the initial model produced a poor model fit to the data (Table 2, Model-1). The values did not meet the recommended range for all the fit indices. Thus, we re-specified the model by removing some non-significant paths such as, POC to PA, Psychological condition to POC, and Mastery to Pros, iteratively. Removal of non-significant pathways were stopped until all pathways in the model were significant to obtain modified model-2. Table 2 shows the model fit indices for the modified model. Based on the results, the fit indices were improved but were still not within the acceptable range values (Table 2, Model-2).

In the SEM output, pathways recommended by modification index (MI) were examined. Several additional pathways were included in the model after adequate theoretical support was carried out by the researchers. The additional pathways were POC to pros, ESE to POC, pros to POC, others' expectations to PA, and psychological condition to PA. Then, the final model (Model-3) was obtained, with a majority of the fit indices achieving the recommended cut-off values (see Table 2, Model-3).

Table 3 presents the detail SEM results of the path relationships in the final model. The table shows all the regression coefficient values of each significant hypothesized pathways with their p-values to display clearer view of the final model as displayed in Fig 2.

The coefficient of determination ($R^2$), the amount of variance in each dependent variable that was explained by the model, ranged from 0.142 to 0.838. The model is illustrated in a diagram in Fig 2.

## Indirect relationships in the final model

Six variables (POC, ESE, physical condition, psychological condition, mastery, and enjoyment) exhibited indirect and positive statistically significant effects on amount of PA. Detailed results of the indirect relationships within the final model are presented in Table 4.

## Discussion

The main objective of the present study was to examine the inter-relationship between the psychological constructs in the TTM with motives for PA and amount of PA. The second

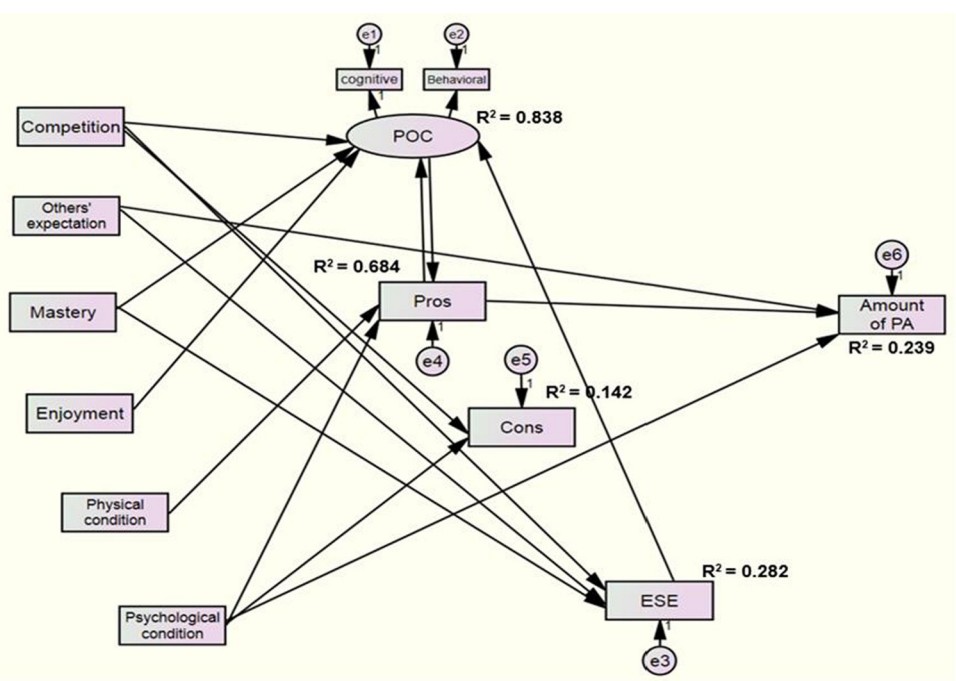

**Fig 2. Final model for the SEM of TTM, motives for PA, and amount of PA.**

objective was to determine any indirect relationship between them. These objectives were achieved through the development of an initial set of specific hypotheses based on the conceptual framework with theoretical and literature support [24, 40]. Significant path relationships between TTM psychological constructs, motives for PA, and amount of PA were determined, as we aimed to understand and assess the psychological factors of behavioral changes and aspects that motivate people with T2DM to perform PA. The structural equation model was developed through the involvement of the POC-M scale, DB-M scale, ESE-M scale, PALMS-M scale, and IPAQ-M, which adopted the TTM psychological constructs, motives for PA, and

**Table 4. Standardized indirect and p-value of the motives for PA and TTM's variables toward amount of PA.**

| Predictor variables | Through | Indirect coefficient value (*p*-value) |
|---|---|---|
| **POC to PA:** | | |
| POC | Pros | 0.105 (0.008) |
| **ESE to PA:** | | |
| ESE | POC, Pros | 0.040 (0.014) |
| **Physical condition to PA:** | | |
| Physical condition | Pros | 0.049 (0.033) |
| **Psychological condition to PA:** | | |
| Psychological condition | Pros | 0.054 (0.036) |
| **Mastery to PA:** | | |
| Mastery | ESE, POC, Pros | 0.014 (0.030) |
| **Enjoyment to PA:** | | |
| Enjoyment | POC, Pros | 0.032 (0.021) |

Notes: POC = processes of change, ESE = exercise self-efficacy, PA = amount of physical activity

amount of PA as measurement models. The pros and cons of DB were treated as separate single factors, as the two contradict each other. In overall, 23% of variance in amount of PA could be explained by motives for PA (others' expectation and psychological condition) and decisional balance (Pros).

From the initial structural equation model, we proposed that all motives of PALMS could affect the POC, pros, cons, and ESE. This implies that any aspects of motivation could improve the manner of participants' behavioral changes, decision making, and their efforts to perform PA. We also proposed that POC, pros and cons, and ESE may affect amounts of PA directly. With positive changes in participants' behavior and positive judgment when making decisions, participants could be more active and may change from a physically inactive to an active lifestyle.

Based on the findings of the SEM analysis, only the pros of DB from the TTM had a direct significant relationship with amounts of PA. This indicates that participants were aware of the benefits of PA for their health condition. They were more focused on the benefits of PA rather than its costs. This result was supported by Burkholder and Nigg [41], who suggested that people in the preparation, action, or maintenance stages are more interested in and attracted to the positive benefits of PA instead of its low-risk negative effects.

The final model also showed a reciprocal relationship between POC and pros. Studies have applied the TTM and yielded reciprocal results between the psychological constructs within the TTM. A study applying an integrated model consisting of five constructs, 1. perceived risk for AIDS, 2. decisional balance, 3. exercise self-efficacy for condom use, 4. lifetime number of sex partners, and 5. behavior risk, was carried out among women at risk for heterosexual transmission of HIV in New England [42]. The results of the study showed reciprocal relations between decisional balance and the exercise of self-efficacy for condom use, suggesting vagueness in the causal influences. Burkholder and Harlow [42] implied the possibility of a natural reciprocal relation between both constructs (i.e., efficacy of condom use may lead to valuation of the pros and cons of condom use which, indirectly, could improve exercise self-efficacy).

Another study that presented reciprocal relation results was performed by Maddison and Prapavessis [43], who tried to examine the potential of variables in the TTM to appear as predictors rather than as consequences of exercise behavior. The participants consisted of students ($n$ = 1434) recruited from 13 high schools in New Zealand. In general, the study results supported a reciprocal type of relationship between most of the examined TTM psychological constructs [43]. A total of three reciprocal relationships were identified in the study. First, the study determined a reciprocal relationship between exercise self-efficacy and exercise behavior, indicating exercise self-efficacy as a crucial cause and effect of exercise. Second, the study determined a reciprocal relationship between the pros of decisional balance and exercise behavior. A reciprocal relationship between counterconditioning and self-liberation of processes of change and exercise behavior was the final reciprocal relation displayed in the study.

As for motives in PALMS, two motives had direct relationships with amount of PA. Others' expectations and psychological conditions were the motives that directly affected participants' amount of PA, based on the final model. Family, friends, and doctors treating them should be the groups to whom they wanted to show an improvement in their health status, as those groups were the closest groups and provided good support. Similar results were found in a study performed by Ferrand et al. [44], who aimed to assess motives that provided patients with T2DM with the strength and spirit to perform regular PA. Ferrand et al. [44] yielded results that showed that medico-sporting educators, peers, and family members helped the participants be in a supportive environment while they were trying to be consistently active. Furthermore, the positive surroundings encouraged participants' psychological states to be more competent and increased their self-determination [44]. The same results were observed

in the present study, where the psychological conditions of the participants directly affected their amount of PA. Desires to improve self-esteem, mental toughness, resilience, and coping skills could be the psychological aspects that motivated participants to be more active. Thus, persistent support from surrounding people (family and friends) with constant counseling from doctors or other medical staff could be great motivators for people with T2DM to stay physically active [45].

Other than a direct significant relationship within the TTM psychological constructs and amount of PA, there was also indirect effect between POC and amount of PA. Through positive decision making (pros), participants' POCs significantly affected the amount of PA. The result was similar to the study involving young Mexican-American women who became more active with the utilization of POC for exercise behavior change [46]. Amount of PA was also affected by the ESE through POC then pros. The participants in the present study possessed good exercise self-efficacy, which helped them go through the cognitive and behavioral POC. Good results from the covert and overt behavior helped the participants make positive decisions to improve their PA performance after learning all the PA benefits for their current health condition. A study that compared the influence of exercise self-efficacy, social support, and perceived barriers on physically inactive patients with T2DM presented similar results on the relation of ESE and PA level [47]. Participants' ESEs were significantly associated with their PA levels, as Adeniyi et al. [47] found that participants with low PA levels possessed low ESE levels.

Three motives of PALMS produced an indirect significant effect on participants' amount of PA: physical condition, mastery, and enjoyment. All of them affected amount of PA through positive decision making (pros) after they evaluated the benefits of PA on blood sugar control. Nonetheless, mastery had ESE and POC, and enjoyment had POC as extra paths before the pros to provide a positive effect on the amount of PA. Physical condition is one of the crucial parts in determining body fitness level or physical fitness level [48]. The indirect relationship of physical condition and PA level among people with T2DM in the present study was supported by previous studies [49–51] that labeled physical fitness level as one of the robust predictors for the T2DM population. Mastery as one of the indirect motivators for participants of the present study to increase PA level was supported by a study done by Raaijmakers et al. [52]. They yielded a positive correlation between diabetic patients' mastery and self-management skills. The ability to manage symptoms, such as increased PA levels, is one of the self-management skills [53]. As for enjoyment, given that the present study participants were in the elderly group age, we found that enjoyment was one of the strong motivators for the elderly to embrace PA [54].

Rashidi Jahan et al. [55] conducted a study to determine the relationship between the TTM components and PA among municipality councils' members in Tehran. Based on the regression test results, the stage of change was the only TTM component that could predict the PA performance of the participants. POC and ESE were the measured TTM components in the study that showed a non-significant result in the regression test. The failure of POC to predict PA level was also supported by Xiong et al. [56], who investigated the differences in PA levels and TTM components among college students in China. From a different perspective, instead of an increase in PA level, POC could also produce a suppressor effect on PA [57–59]. Hence, the non-significant direct effect of POC on amount of PA should be expected in TTM and PA relationship research.

Liu et al. [24] showed that ESE had a non-significant and direct effect on amount of PA. The final structural equation model of the study showed that POC was the only TTM component that directly affected amount of PA among the university students. However, the pros had a significant indirect effect on amount of PA through POC [40], which was contrary to the SEM results of the present study. As for the cons, the variable was commonly found to not

have a significant effect on amount of PA through several studies of the relationship between TTM components and PA [39, 40, 56, 60].

From the final model, it shows that participants tend to be more active after taking into account all the benefits of PA for their health status. Psychological condition and others' expectation were seeming to be able to give more motivation for participants performing PA compared to the other motives of PALMS. As for indirect relationship pathway to amount of PA, POC and ESE of TTM were also improve participants' amount of PA. However, participants' POC and ESE required evaluation of PA benefits to initiate the exercise behavior change processes and rising up their self-efficacy before they become more physically active. Same goes for Physical condition, Mastery, and Enjoyment that have indirect significant relationship with amount of PA, yet need through the positive decisional-making from the Pros factor of DB. As an overall result of the SEM analysis, we found that although not all of the TTM psychological constructs and motives for PA were significantly related and may affect the participants' amount of PA, yet most of the constructs were proven to be effective to improve amount of PA among Malaysian with T2DM.

There were some limitations in the present study. Generalizability of the findings is the primary limitation as all of the participants were recruited only in a single hospital in Kelantan and only focusing on people who suffering from T2DM. Other limitation is the quantitative methods that need to be used to measures the variables among participants. Respondent bias, acquiescence bias, demand characteristics, extreme responding, and social desirability bias could happen as the participants could answer the questionnaire dishonestly and insincerely. These biases may give a negative outcome on the reliability of the instruments/questionnaire and could be a serious concern of the study. However, during the data collection, the researcher has encouraged the participants to answer the questionnaire sincerely and only according to what they were think and their current condition.

The current study has provided useful information to the health care providers, people with T2DM and their family about the significant factors that could increase amount of PA. This information could be used for health care planners and diabetes educators in providing better treatment and intervention to people with T2DM in enhancing their PA level. Despites the presented positive results, better results could be possibly obtained with some improvement and adjustment in the future studies. The present study was carried out only in a single hospital in Kelantan. With larger setting and research support, multi-center cross-sectional study could be done by involving more hospitals in Kelantan and/or in Malaysia. The multi-center study could recruit balanced multiracial participants and wider range of age group. More generalizability results could be produced through larger setting with bigger sample size.

## Conclusion

A structural equation model between the TTM psychological constructs, motives for PA, and amount of PA was successfully developed. From the final model, 16 paths were statistically significant. There were some direct effects between motives for PA to POC, DB, ESE and amount of PA. There were also indirect effects observed among the TTM's variables and motives for PA toward the amount of PA. These findings could be beneficial for health educators and health care providers in improving the PA level for people with T2DM.

## Supporting information

**S1 Data. SEM data.**
(PDF)

## Acknowledgments

We would like to give a great acknowledgement to all the participants who willingly and voluntarily provided their commitment and cooperation in this study. We also thank the Director of Hospital USM, who gave us permission to perform data collection, and the hospital staff who helped us during the data collection.

## Author Contributions

**Conceptualization:** Aizuddin Hidrus, Yee Cheng Kueh, Bachok Norsa'adah, Garry Kuan.

**Formal analysis:** Aizuddin Hidrus, Yee Cheng Kueh.

**Funding acquisition:** Yee Cheng Kueh.

**Investigation:** Aizuddin Hidrus, Yee Cheng Kueh, Garry Kuan.

**Methodology:** Aizuddin Hidrus, Yee Cheng Kueh, Bachok Norsa'adah, Garry Kuan.

**Project administration:** Aizuddin Hidrus.

**Software:** Yee Cheng Kueh.

**Supervision:** Yee Cheng Kueh, Bachok Norsa'adah, Garry Kuan.

**Writing – original draft:** Aizuddin Hidrus, Yee Cheng Kueh.

**Writing – review & editing:** Aizuddin Hidrus, Yee Cheng Kueh, Bachok Norsa'adah, YoungHo Kim, Yu-Kai Chang, Garry Kuan.

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
