## [Decision Letter · Decision Letter 0]

13 Sep 2021

PONE-D-21-23408Structural Equation Model of Psychological Constructs of Transtheoretical Model, Motives for Physical Activity, and Physical Activity Amount in People with Type 2 Diabetes MellitusPLOS ONE

Dear Dr. Kueh,

Thank you for submitting your manuscript to PLOS ONE. After careful consideration, we feel that it has merit but does not fully meet PLOS ONE’s publication criteria as it currently stands. Therefore, we invite you to submit a revised version of the manuscript that addresses the points raised during the review process.

We look forward to receiving your revised manuscript.

Kind regards,

Hoh Boon-Peng, PhD

Academic Editor

PLOS ONE

Journal Requirements:

Reviewers' comments:

Reviewer's Responses to Questions

**Comments to the Author**

1. Is the manuscript technically sound, and do the data support the conclusions?

Reviewer #1: Yes

Reviewer #2: Yes

2. Has the statistical analysis been performed appropriately and rigorously? 

Reviewer #1: Yes

Reviewer #2: Yes

3. Have the authors made all data underlying the findings in their manuscript fully available?

Reviewer #1: Yes

Reviewer #2: Yes

4. Is the manuscript presented in an intelligible fashion and written in standard English?

Reviewer #1: Yes

Reviewer #2: Yes

5. Review Comments to the Author

Reviewer #1: Minor revision. Kindly refer to the attachment for suggestion. Thanks.

Reviewer #2: Thank you dear authors for the good work. The research problem is relevant, and testing of such models is so important for practitioners to consider improvements in their intervention. Based on my review i put the following specific comments.

Title:Include the place of study(Malaysia)

1. Abstract: the study design used was Mixed method, so the term cross sectional and purpose sampling are not important. sample size and sampling method for both quantitative and qualitative design need to be presented in the methods section.

2. Introduction: Psychologically, the transtheoretical model (TTM) is a questionnaire...(TTM is a behavior change model not a questionnaire). The research problem/gaps is should be described more; i.e Physical Activity in People with Type 2 Diabetes Mellitus in Malaysia or the world. what is indirect relationship and why it is needed?

3. Materials and Methods; the study design section needs clarity. sampling method and participant section or sampling procedure are not clear.describe each study designs used, how participants were chosen, how many etc. take the statement "All participants were initially required

to complete consent forms before the study began as an indication that they volunteered and were

willing to participate in this study" to ethical consideration section. the Measures and Materials is nice.from whom you collect the data? patient them selves or their medical card. some relevant variables such as religion are missed. if you used cards, it secondary data and you should clearly mention that you used secondary data and its limitations.

4. Results: the last two variables in table one are not demographic characteristics. Rather, they are clinical. the title of tables need to be written properly. what where and when for each table. the final model( the relationship between the pathways and the PA) should be interpreted in detail. This is the core of your research and same is true in discussion section. You use only a cross sectional quantitative study. at the beginning it seems you used a mixed method. In general it is a good work. there are some spelling errors like the word University in many sections. with careful revision, it will be a good article.

6. PLOS authors have the option to publish the peer review history of their article (what does this mean?). If published, this will include your full peer review and any attached files.

Reviewer #1: No

Reviewer #2: No

---

## [Author Response · Author response to Decision Letter 0]

26 Sep 2021

Reviewer #1: 

1. What is the meaning of “Psychological Constructs” representing in the present study

Response: “Psychological Constructs” in the present study represent the psychological variables of the Transtheoretical Model which include, Processes of change, Decisional balance, and Exercise Self-efficacy. See line 78-82.

2. Lack of rationale to support the model. SEM use to test the model not to develop model. Therefore, need citation/theories/model to support the indirect/mediation variable/structure of the model

Response: We thank the reviewer for the suggestion. We added a citation in the last paragraph of Introduction section regarding rationale to support the model. See line 99-103. 

a. Why TTM as mediator (even mediation not in the analysis but the present paper focus on indirect effect)?

Response: The indirect relationship is part of the SEM analysis. It is not the focus of the paper. But it provides additional information about the indirect relationship of motives for PA and psychological constructs on the amount of PA. If the indirect relationship is significant, it indicates that there is a significant mediator in the model. 

b. Kindly present and report direct effect when reported indirect effect

Response: We thank the reviewer for the suggestion. Direct effects are reported in Table 3 (standardised regression coefficient). Thus, we do not repeat the direct effects results in Table 4. 

c. Kindly present the outcome of measurement model in brief.

Response: We thank the reviewer for the suggestion. Outcome of measurement model presented in section “Measures and Materials” as we adopted the translated Malay version of all the scales from the previous validation studies (see line 124-183). Thus, the measurement model is not described in the present paper. 

d. Three model reported and final Model 3 used according to the model fit values. However, which item(s) deleted or not fulfil the mode requirement. Too brief and not clear, kindly enhance.

Response: We have now added the removed pathways from the model 1 to obtain model 2 in the first paragraph of Structural Model subsection. Additional pathways based on the modification index recommendation were presented in the second paragraph.

e. Table 3 present the significant path, how about the non-significant path(s). The path deleted from the model during model fit or during the measurement model analysis or during the structural model analysis?

Response: We thank the reviewer for the comment. The path deleted from the model during the structural model analysis as stated in subsection Structural Model.

f. How the value of indirect effect calculates? Where the significant value from?

Response: The value of indirect effects and its p-values were obtained from the Mplus output. It is part of the SEM analysis. We have futher explained this in line 222.

3. What the age (years) of the participants? Kindly report in group, not mean.

Response: We thank the reviewer for the suggestion. We have now added age group of the participants in Table 1.

4. Discussion too brief and more focus on reported previous studies. Discussion should align with the purpose of the study.

 a. What the overall contribution? -23% refer to good or moderate etc?

Response: The overall R2 on amount of PA indicated that 23% of variance in amount of PA could be explained by PA motives (others’ expectation and psychological condition) and decisional balance (Pros). We do not interpret the 23% as good or moderate, because the interpretation is very depend on the field of study. There is no gold standard available in the literature to label the level of coefficient of determination (R2). We report the values and allowed readers to make their own judgement. 

b. What the contribution of indirect effect presented in this model outcome? What’s the value or impact of the findings?

Response: We have now added a paragraph in the discussion to explain on the value or impact of the findings on the participants. See line 389-400.

5. No implication and further research suggestion. Kindly revise and enchance the Discussion.

Response: We thank the reviewer for the suggetion. We have now added a paragraph (last) in the Discussion section to discuss future studies recommendations. 

Reviewer #2: 

1. Title: Include the place of study

Response:

We thank the reviewer for the suggestion. We have added place of study (Malaysia) in the title.

2. Abstract: the study design used was Mixed method, so the term cross sectional and purpose sampling are not important. sample size and sampling method for both quantitative and qualitative design need to be presented in the methods section.

Response: We thank the reviewer for the comment. We rephrased the sentence to indicate that was only a cross-sectional study design and non-probability sampling method purposive sampling was used. We also added the sample size of the present study in the method section.

3. Introduction: Psychologically, the transtheoretical model (TTM) is a questionnaire...(TTM is a behavior change model not a questionnaire). 

Response: We have now rephrased the the sentence in the third paragraph.

4. Materials and Methods; the study design section needs clarity. sampling method and participant section or sampling procedure are not clear.describe each study designs used, how participants were chosen, how many etc. take the statement "All participants were initially required to complete consent forms before the study began as an indication that they volunteered and were willing to participate in this study" to ethical consideration section. the Measures and Materials is nice.from whom you collect the data? patient them selves or their medical card. some relevant variables such as religion are missed. if you used cards, it secondary data and you should clearly mention that you used secondary data and its limitations.

Response: We thank the reviewer for the comment. We added clarification in the first paragraph of Study design section regarding the cross-sectional study design and non-randomized sampling for this study. The inclusion and exclusion criteria of participants are included in the same paragraph. We also removed the statement "All participants were initially required to complete consent forms before the study began as an indication that they volunteered and were willing to participate in this study" to ethical consideration section. The data was collected from the patient themselves, not from their medical card. We did not collect their religion variables.

5. Results: the last two variables in table one are not demographic characteristics. Rather, they are clinical. the title of tables need to be written properly. what where and when for each table. the final model( the relationship between the pathways and the PA) should be interpreted in detail. This is the core of your research and same is true in discussion section. You use only a cross sectional quantitative study. at the beginning it seems you used a mixed method. In general it is a good work. there are some spelling errors like the word University in many sections. with careful revision, it will be a good article.

Response: We thank for the comment. The title of the tables have been revised properly. The final model has been further interpreted in detail in the Discussion section.

---

## [Decision Letter · Decision Letter 1]

28 Dec 2021

PONE-D-21-23408R1Structural Equation Model of Psychological Constructs of Transtheoretical Model, Motives for Physical Activity, and Amount of Physical Activity among People with Type 2 Diabetes Mellitus in MalaysiaPLOS ONE

Dear Dr. Kueh,

Thank you for submitting your manuscript to PLOS ONE. After careful consideration, we feel that it has merit but does not fully meet PLOS ONE’s publication criteria as it currently stands. Therefore, we invite you to submit a revised version of the manuscript that addresses the points raised during the review process.

We look forward to receiving your revised manuscript.

Kind regards,

Hoh Boon-Peng, PhD

Academic Editor

PLOS ONE

Journal Requirements:

Reviewers' comments:

Reviewer's Responses to Questions

**Comments to the Author**

1. If the authors have adequately addressed your comments raised in a previous round of review and you feel that this manuscript is now acceptable for publication, you may indicate that here to bypass the “Comments to the Author” section, enter your conflict of interest statement in the “Confidential to Editor” section, and submit your "Accept" recommendation.

Reviewer #1: All comments have been addressed

Reviewer #2: All comments have been addressed

2. Is the manuscript technically sound, and do the data support the conclusions?

Reviewer #1: Yes

Reviewer #2: Partly

3. Has the statistical analysis been performed appropriately and rigorously? 

Reviewer #1: Yes

Reviewer #2: Yes

4. Have the authors made all data underlying the findings in their manuscript fully available?

Reviewer #1: Yes

Reviewer #2: Yes

5. Is the manuscript presented in an intelligible fashion and written in standard English?

Reviewer #1: Yes

Reviewer #2: Yes

6. Review Comments to the Author

Reviewer #1: All comments have been addressed and explained. Author(s) amended and improved the manuscript with justification. Suggest for formatting.

Reviewer #2: Thank you dear authors, i realize that most of the comments are corrected. However, i have still concern on the sample size determination and sample section. You need to show how you come up with 331 participants/calculate showing the formula and the parameters used and how you chose them. I am not convinced about the selection of your study participants. In addition, include how you collected the data and the quality control measure used .

7. PLOS authors have the option to publish the peer review history of their article (what does this mean?). If published, this will include your full peer review and any attached files.

Reviewer #1: No

Reviewer #2: No

---

## [Author Response · Author response to Decision Letter 1]

1 Mar 2022

Reviewer 2:

1. Thank you dear authors, i realize that most of the comments are corrected. However, i have still concern on the sample size determination and sample section. You need to show how you come up with 331 participants/calculate showing the formula and the parameters used and how you chose them. I am not convinced about the selection of your study participants. In addition, include how you collected the data and the quality control measure used .

Response: We thank you for the comment. We have added the sample size calculation using monte carlo simulation in Mplus software. New subsection for this part. Also, we added new subsection explaining on how data collection and participants’ recruitment were performed. References also revised according to the additional citation in the sample size calculation subsection.

---

## [Decision Letter · Decision Letter 2]

15 Mar 2022

Structural Equation Model of Psychological Constructs of Transtheoretical Model, Motives for Physical Activity, and Amount of Physical Activity among People with Type 2 Diabetes Mellitus in Malaysia

PONE-D-21-23408R2

Dear Dr. Kueh,

We’re pleased to inform you that your manuscript has been judged scientifically suitable for publication and will be formally accepted for publication once it meets all outstanding technical requirements.

Kind regards,

Hoh Boon-Peng, PhD

Academic Editor

PLOS ONE

Additional Editor Comments (optional):

Reviewers' comments:

Reviewer's Responses to Questions

**Comments to the Author**

1. If the authors have adequately addressed your comments raised in a previous round of review and you feel that this manuscript is now acceptable for publication, you may indicate that here to bypass the “Comments to the Author” section, enter your conflict of interest statement in the “Confidential to Editor” section, and submit your "Accept" recommendation.

Reviewer #1: All comments have been addressed

Reviewer #2: All comments have been addressed

2. Is the manuscript technically sound, and do the data support the conclusions?

Reviewer #1: Yes

Reviewer #2: Yes

3. Has the statistical analysis been performed appropriately and rigorously? 

Reviewer #1: Yes

Reviewer #2: Yes

4. Have the authors made all data underlying the findings in their manuscript fully available?

Reviewer #1: Yes

Reviewer #2: Yes

5. Is the manuscript presented in an intelligible fashion and written in standard English?

Reviewer #1: Yes

Reviewer #2: Yes

6. Review Comments to the Author

Reviewer #1: All comments have been addressed and amended. Article improve and enhance accordingly.

Suggest for acceptance

Reviewer #2: I have still some comments to be considered by the authors.

1. In the introduction of the abstract, before you state the objective , it is better to put a ststement about the research gap. you can take form the last paragraph of the intoruction.

2. I am not convinced why participants are selected purposively? What was the purpose of seclection? You can interview all patients visiting the during your data collection period and untill you get the calculated sample size.

3. table 1 ; the variables presented are not only demographic. example; Hg A1C.BMI

With careful revision, i accept it

7. PLOS authors have the option to publish the peer review history of their article (what does this mean?). If published, this will include your full peer review and any attached files.

Reviewer #1: No

Reviewer #2: No

---

## [Editor Report · Acceptance letter]

23 Mar 2022

PONE-D-21-23408R2 

Structural Equation Model of Psychological Constructs of Transtheoretical Model, Motives for Physical Activity, and Amount of Physical Activity among People with Type 2 Diabetes Mellitus in Malaysia 

Dear Dr. Kueh:

I'm pleased to inform you that your manuscript has been deemed suitable for publication in PLOS ONE. Congratulations! Your manuscript is now with our production department. 

Kind regards, 

on behalf of

Dr. Hoh Boon-Peng 

Academic Editor

PLOS ONE